# Pt_3_Ni@C Composite Material Designed and Prepared Based on Volcanic Catalytic Curve and Its High-Performance Static Lithium Polysulfide Semiliquid Battery

**DOI:** 10.3390/nano11123416

**Published:** 2021-12-16

**Authors:** Ying Wang, Yao Yao, Yu Chen, Jiyue Hou, Zhicong Ni, Yanjie Wang, Xiuqiong Hu, Yanzhong Sun, Rui Ai, Yulin Xian, Yiyong Zhang, Xue Li, Yingjie Zhang, Jinbao Zhao

**Affiliations:** 1National and Local Joint Engineering Laboratory for Lithium-Ion Batteries and Materials Preparation Technology, Faculty of Metallurgical and Energy Engineering, Kunming University of Science and Technology, Kunming 650093, China; wy0918wy@163.com (Y.W.); yyao310@stu.kust.edu.cn (Y.Y.); hgeneral_jy@163.com (J.H.); ni_zhicong@163.com (Z.N.); 18298325286@163.com (Y.W.); 18213874923@139.com (Y.X.); zhangyingjie09@126.com (Y.Z.); 2College of Electrical Information Engineering, Panzhihua University, Panzhihua 617000, China; huxiuqiongdu@163.com; 3State Key Laboratory of Physical Chemistry of Solid Surfaces, State-Province Joint Engineering Laboratory of Power Source Technology for New Energy Vehicle, Collaborative Innovation Center of Chemistry for Energy Materials, College of Chemistry and Chemical Engineering, Xiamen University, Xiamen 361005, China; 20620191151307@stu.xmu.edu.cn (Y.C.); syzxdm@163.com (Y.S.); jbzhao@xmu.edu.cn (J.Z.); 4College of Vanadium and Titanium, Panzhihua University, Panzhihua 617000, China; AR13281506559@163.com

**Keywords:** Pt_3_Ni@C, lithium polysulfide semiliquid battery, volcanic catalytic curve, shuttle effect

## Abstract

There are many challenges for the Static lithium polysulfide semiliquid battery in its commercial application, such as poor stability of the cathode material and further amplification of the lithium polysulfide shuttle effect. Therefore, this manuscript introduced a new type of Pt_3_Ni@C composite material as cathode working electrode based on the principle of volcanic catalytic curve. Through symmetric battery test, CV, polarization curves and impedance test, it was found that Pt_3_Ni@C composite material had good catalytic activity of lithium polysulfide to improve electrochemical kinetics. When the catholyte was Li_2_S_8_ and the charge-discharge voltage range was 1.8~2.6 V, the capacity maintained at approximately 550 mAh g^−1^, and the coulombic efficiency maintained at approximately 95% after 100 cycles at a current rate of 0.5 mA cm^−2^. The Pt_3_Ni@C composite material is a potential cathode material with the specific capacity and long cycling stability of the static lithium polysulfide semiliquid battery.

## 1. Introduction

In recent decades, solar energy, tidal energy, wind energy and other renewable energies have emerged and made remarkable progress. However, due to the impact of the external environment, these energy sources are intermittent and unstable. Therefore, it is imperative to develop a large-scale energy storage system with high performance, low cost, environmental friendliness and safety [1,2,3,4,5]. Lithium–sulfur batteries (LSBs) were one of the promising candidates in the next-generation battery system, and their theoretical energy density could reach 2600 Wh kg^−1^ [6,7]. Among them, the sulfur cathode had many advantages containing a high specific capacity (the theoretical specific capacity of elemental sulfur was as high as 1675 mAh g^−1^), being rich in sulfur, low in price, and environmentally friendly [8,9]. However, the infamous ‘shuttle effect’ of lithium polysulfide caused several problems, such as poor battery cycle performance, fast capacity degradation and overcharge [10]. This severely hindered the further development and commercial application of LSBs.

Among all kinds of electric energy storage technologies, flow batteries have attracted extensive attention of researchers due to their long cycle life (about 800–3000 cycles for lithium–ion battery, and 10,000 cycles for flow batteries), modular design of battery mold and controllable energy density. The active substances of the anode or cathode are stored in the liquid storage tank, and the energy is stored through the circulation pumps, which makes it flexible to assemble and use [11,12,13,14,15]. Nevertheless, traditional flow batteries (vanadium, zinc and organic flow batteries) [11,12,14,15] have many problems such as low operating voltage (about 1.5 V, narrow potential window of aqueous electrolyte), and limited solubility of active substances in the electrolyte (approximately 1.5 M) [16], resulting in low energy densities. In order to optimize the energy density and decrease the system cost, new active substances and chemical processes are being explored [17,18,19,20,21,22,23,24,25,26,27]. When the organic solution is selected as the electrolyte carrier, the redox active substances with lower potential and higher solubility can be selected in the flow batteries, which can produce higher working voltage. However, active organic substances have been widely studied [28,29,30,31,32,33]. The concentration of redox molecules in the organic electrolyte is low, which further limits the increase of energy density.

Many studies were reported on LSBs indicating that lithium polysulfide can be easily dissolved in organic electrolytes with a solubility of up to 7 M [16,34]. Therefore, the development of the lithium polysulfide semiliquid battery was proposed using lithium metal as the anode and lithium polysulfide organic solution as the catholyte equipped with high energy density at low cost. However, the insufficient capacity achievement issue was mainly correlated with the shuttle effect and the bad kinetics of the polysulfide redox reaction. The conversion kinetics between lithium polysulfide is slow, especially the conversion between solid Li_2_S/Li_2_S_2_ and other lithium polysulfide. Therefore, it is very important to find a cathode working electrode material that can enhance the interaction force with lithium polysulfide and promote the conversion between lithium polysulfide. In recent years, the concept of electrocatalytic polysulfide redox in the field of LSBs was proposed. It was the matrix material that could promote electrochemical reaction and charge transfer while adsorbing polysulfide, and ultimately improve the energy efficiency of LSBs, capacity retention and cycle stability [35,36]. Arava et al. [37] found that metal materials such as Pt and Ni could effectively promote the reaction kinetics of polysulfides, further increasing their battery capacity.

Thus, according to the catalyst volcanic curve principle [29], this manuscript designed and prepared Pt_3_Ni@C composite material as a cathode working electrode to promote the conversion of lithium polysulfide, thereby improving the electrochemical performance of the static lithium polysulfide semiliquid battery. Pt_3_Ni@C composite material together with a 0.5 M Li_2_S_8_ catholyte as the positive electrode area, lithium metal anode and lithium sulfur electrolyte as the negative electrode area, which help to inhibit the polysulfide shuttle effect, increase the utilization of the active lithium polysulfide, and boost the catalytic conversion of lithium polysulfide.

## 2. Experimental Section

### 2.1. Preparation of Pt_3_Ni@C Composite Materials

Firstly, 40 mg of Super P was added to a 50 mL round bottom flask containing 20 mL of dimethylformamide (DMF), and sonicated for 1 h to obtain a carbon carrier dispersion. Then, 16 mg of platinum diacetylacetonate (Pt(acac)_2_), 8 mg of nickel diacetylacetonate (Ni(acac)_2_) and 122 mg of benzoic acid (C_6_H_5_COOH) were added into the flask and proceed ultrasonically treatment for 30 min. After that, the resulting mixed dispersion was heated in a constant temperature water bath at 160 °C and reacted for 24 h. The resulting product was centrifuged at 1100 rpm for 20 min, washed with ethanol 3 times, and dried at 60 °C to obtain Pt_3_Ni@C composite material.

### 2.2. Material Characterization

The morphology of the sample was studied by transmission electron microscope (TEM, JEM-2100 (JEOL, Tokyo, Japan)) at 200 kV. The structure of the product was characterized by X-ray diffractometer (the Philips X’pert Pro Super X-ray diffract meter and Cu Ka radiation) at 60 kV, 60 mA, scanning range 10–90° and scanning speed 5° min^−1^.

### 2.3. Cell Assembly and Electrochemical Measurements

Firstly, the active material (the Pt_3_Ni@C composite material), conductive agent (Super P) and adhesive (polyvinylidene fluoride (PVDF)) were accurately weighed at a mass ratio of 8:1:1. N-methylpyrrolidone (NMP) was used as a dispersant, and the mixed powder was magnetic stirred for 2 h to obtain active material slurry, which was coated on a stainless steel mesh to obtain a working electrode. The thickness of the slurry coating was 100 microns. Finally, the solvent was removed by vacuum drying at 60 ℃ for 12 h to obtain a cathode working electrode. The cathode working electrode and the lithium polysulfide catholyte were used as the positive electrode area, and the lithium metal anode and the lithium sulfur electrolyte were used as the negative electrode area. In a glove box with argon atmosphere, the lithium polysulfide semiliquid battery was obtained by stacking the cathode working electrode, the catholyte (the electrolyte contained about 0.5 M Li_2_S_8_ prepared by adding Li_2_S and S (mole ratio of 1:7) into the conventional lithium–sulfur battery electrolyte), PP/PE/PP porous membrane, the lithium sulfur electrolyte (1M lithium trifluoromethanesulfonate (LiTFEST) (DOL/DME = 1:1, V:V)) and the lithium metal anode in sequence.

Using the NEWARE battery test system (Shen Zhen Xinwei New Energy Technology Co., Ltd., Shenzhen, China) for constant current charge and discharge test, the current density was 0.5 mA cm^−2^ and the voltage range was 1.5~3.0 V. The CV test was carried out by electrochemical workstation (CHI 660D). The voltage window from −0.5 V to 0.5 V and scanning rate are 50 mV s^−1^ and 0.1 mV s^−1^, respectively. The AC impedance (EIS) tests were carried out by autolab (Metrohm electrochemical workstation) at room temperature. The AC impedance was tested under the conditions of an amplitude of 5 mV and a frequency of 0.01 Hz~100 kHz.

## 3. Results and Discussion

Easily soluble in the electrolyte is the shortcoming of lithium polysulfide in LSBs that can be transformed into an advantage, and a static lithium polysulfide semi-liquid battery is designed and constructed. The schematic diagram of the structure is shown in Figure 1a. The static lithium polysulfide semiliquid battery is composed of a positive electrode area, a negative electrode area and a separator. The positive electrode area includes a cathode working electrode and the lithium polysulfide catholyte. The cathode working electrode is composed of active material (Pt_3_Ni@C composite material), a conductive agent (Super P), binder (PVDF) and current collector (stainless steel mesh). The negative electrode area is composed of a lithium metal anode and lithium–sulfur battery electrolyte. The separator is a PP/PE/PP three-layer porous membrane. (The separator is preferably an ion-selective permeable membrane, allowing only lithium ions to pass through, and further research will continue.) The working principle is as follows: when the static lithium polysulfide semiliquid battery is working, the deposition/stripping reaction of lithium metal anode occurs in the negative electrode area, and the redox conversion reaction of lithium polysulfide occurs in the positive electrode area. Therefore, the static lithium polysulfide semiliquid battery has a high theoretical energy density. In the positive electrode area, the reaction process of the active material lithium polysulfide on the working electrode is shown in Figure 1b. For the battery to work stably, lithium polysulfide must be reversibly and efficiently redox on the cathode working electrode. However, the interconversion between lithium polysulfide is a slow kinetic process, especially the conversion between solid Li_2_S/Li_2_S_2_ and other lithium polysulfide. At the same time, Li_2_S/Li_2_S_2_ are insulators; therefore, the cathode working electrode needs to have both excellent conductivity and electrocatalytic performance. Hence, the Pt_3_Ni@C was designed and prepared as the material of the cathode working electrode, playing the role of conduction and electrocatalysis to promote the efficient conversion of lithium polysulfide.

The morphology of the samples analyzed by a transmission electron microscope (TEM) is shown in Figure 2. Pt_3_Ni particles with a size range from 5 nm to 10 nm are uniformly supported on the Super P, which is beneficial to enhance the conductivity of the composite material, increasing the contact area with lithium polysulfide, and promoting the conversion of lithium polysulfide. As shown in Figure 3, it is the XRD pattern of Pt_3_Ni@C composite material. By comparing the XRD patterns, all the diffraction peaks are consistent with those of Pt_3_Ni@C prepared by Y. Liu et al. [38], indicating that the Pt_3_Ni@C composite material was successfully synthesized.

Figure 4 is the electrochemical performance of the Pt_3_Ni@C-Li_2_S_8_ composite material. It can be seen from Figure 4a,b that the charge and discharge platforms and redox peaks of the static lithium polysulfide semiliquid battery are consistent with those of the traditional LSBs. The lower voltage plateau of the discharge curve in the first cycle is due to the large activation impedance. In the CV, the oxidation peak split, caused by Pt_3_Ni@C, promoted the conversion of lithium polysulfide. The static lithium polysulfide semiliquid battery with Pt_3_Ni@C composite material as the cathode working electrode has a higher specific capacity than C as the cathode working electrode (Figure 4c), which shows that Pt_3_Ni@C has good catalytic activity and can promote the conversion of lithium polysulfide. When the catholyte is Li_2_S_8_ and the charge–discharge voltage range is 1.8~2.6 V, the capacity maintains at approximately 550 mAh g^−1^, and the coulomb efficiency maintains at approximately 95% after 100 cycles at a current rate of 0.5 mA cm^−2^.

In order to verify that the Pt_3_Ni@C composite material can promote the conversion of lithium polysulfide, a symmetric battery test was carried out. It can be seen from Figure 4d that the battery capacitance of Pt_3_Ni@C with Li_2_S_8_ (456 F g^−1^) was greater than that of C with Li_2_S_8_ (370 F g^−1^), which showed that Pt3Ni@C could adsorb more lithium polysulfide and had more reaction sites. The CV measurements of these symmetric batteries were performed between −0.5 V and 0.5 V at a scan rate of 50 mV s^−1^ and 0.1 mV s^−1^, respectably, and the results were shown in Figure 5. Regardless of the fast scan rate of 50 mV s^−1^ (electrochemical polarization control), as shown in Figure 5a, or the slow scan rate of 0.1 mV s^−1^ (concentration polarization control), as shown in Figure 5b, the current responses of Pt_3_Ni@C symmetrical battery are higher than that of the C, demonstrating that Pt_3_Ni@C can facilitate the electrochemical conversion of polysulfide [12,23,37].

The reduction peak of CV corresponds to the conversion of long-chain lithium polysulfide to short chain, and the oxidation peak corresponds to the conversion of short-chain Li_2_S_2_ and Li_2_S to long-chain. Comparing the CV curves of Pt_3_Ni@C-Li_2_S_8_//Li and C-Li_2_S_8_//Li (Figure 6a,b), the peak current of the Pt_3_Ni@C-Li_2_S_8_//Li increased and the peak position of the Pt_3_Ni@C-Li_2_S_8_//Li was shifted positively, which indicated that Pt_3_Ni@C had good catalytic activity for the conversion process of lithium polysulfide. The negative sweep and positive sweep Tafel curves of Pt_3_Ni@C-Li_2_S_8_//Li and C-Li_2_S_8_//Li were shown in Figure 6c,d. It was demonstrated that when Pt_3_Ni@C-Li_2_S_8_//Li and C-Li_2_S_8_//Li were cathodic polarization, the exchange current densities were 190 mA cm^−2^ and 97 mA cm^−2^, respectively. When Pt_3_Ni@C-Li_2_S_8_//Li and C-Li_2_S_8_//Li were anodic polarization, the exchange current densities were 26.9 mA cm^−2^ and 11.0 mA cm^−2^, respectively. Whether it is cathodic polarization or anodic polarization, the exchange current density of Pt_3_Ni@C-Li_2_S_8_//Li more exceeded that of C-Li_2_S_8_//Li, indicating that Pt_3_Ni@C has a strong ability to acquire and lose electrons, thus equipping with a better catalytic activity. 

As shown in Figure 7, it was electrochemical impedance spectroscopy of Pt_3_Ni@C-Li_2_S_8_//Li and C-Li_2_S_8_//Li. The Nyquist plots of Pt_3_Ni@C-Li_2_S_8_//Li and C-Li_2_S_8_//Li before cycling were consisted of a depressed semicircle and an inclined line, corresponding to charge transfer resistance and mass transfer resistance, respectively. The semicircle of Pt_3_Ni@C-Li_2_S_8_//Li was much smaller than that of C-Li_2_S_8_//Li, indicating that the charge transfer resistance of Pt_3_Ni@C-Li_2_S_8_//Li was also relatively lower. It was indicated furthermore Pt_3_Ni@C as the working electrode had better catalytic activity to improve electrochemical kinetics.

## 4. Conclusions

In summary, according to the principle of the volcanic catalytic curve, a new type of Pt_3_Ni@C composite material was successfully prepared to improve electrochemical performance of the static lithium polysulfide semiliquid battery. The Pt_3_Ni@C composite material had a strong interaction force and good catalytic activity for lithium polysulfide, improving the utilization rate and reaction rate of lithium polysulfide. When employed into the static lithium polysulfide semiliquid battery, Pt_3_Ni@C showed high capacity, high coulombic efficiency and long-term cycling stability. After 100 cycles, the capacity maintained at approximately 550 mAh g^−1^, and the Coulomb efficiency maintains at approximately 95%. Pt_3_Ni@C is a potential working electrode material for the development of static lithium polysulfide semiliquid battery.

## Figures and Tables

**Figure 1 nanomaterials-11-03416-f001:**
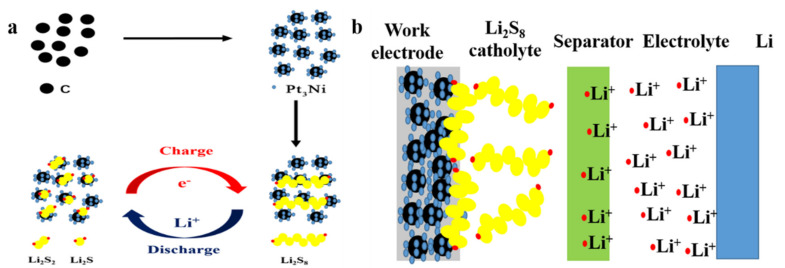
(**a**) The schematic diagram of the static lithium polysulfide semiliquid battery structure. (**b**) In the positive electrode area, the reaction process of the active material lithium polysulfide on the working electrode.

**Figure 2 nanomaterials-11-03416-f002:**
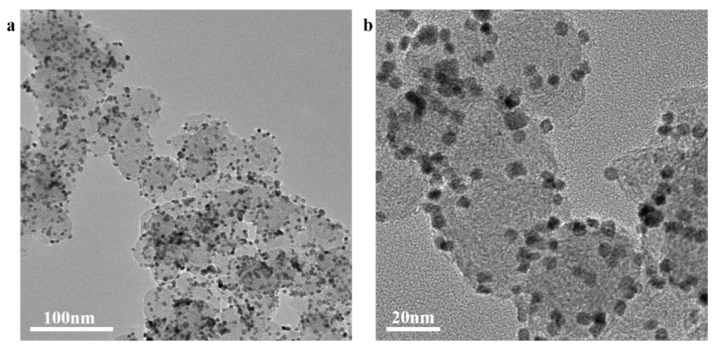
The TEM images of Pt_3_Ni/C composite material: (**a**) magnification is 60,000 times; (**b**) magnification is 200,000 times.

**Figure 3 nanomaterials-11-03416-f003:**
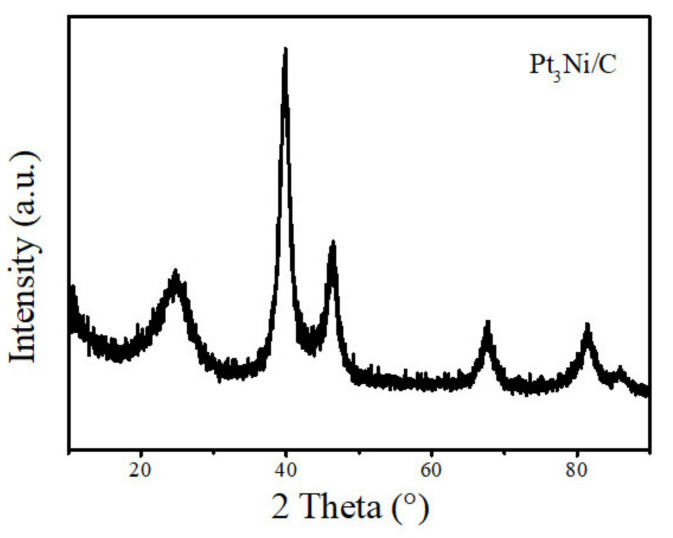
The XRD pattern of Pt_3_Ni@C composite material.

**Figure 4 nanomaterials-11-03416-f004:**
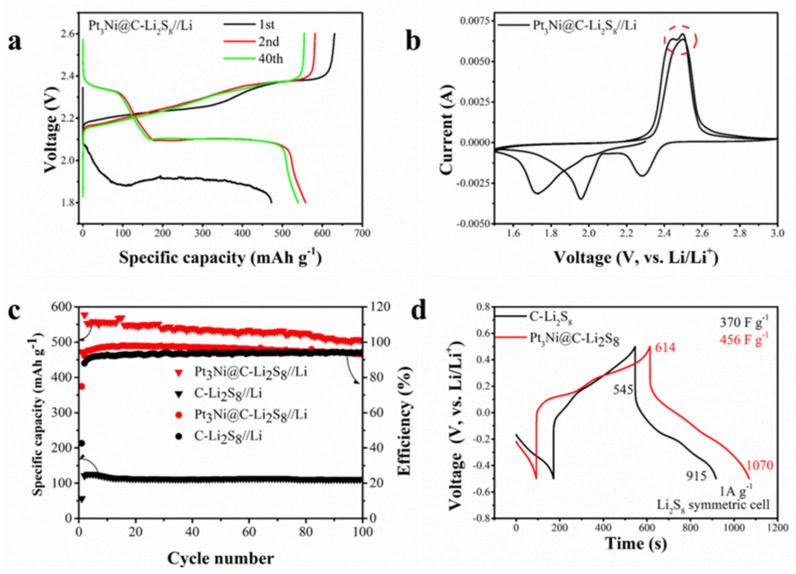
(**a**) Charge and discharge curve. (**b**) Cyclic voltammetry curve (scan rate 0.1 mV s^−1^) of Pt_3_Ni composite material as cathode working electrode. The peak current is marked with a red dotted circle. (**c**) Cycle performance of Pt_3_Ni composite material and C as cathode working electrode. (**d**) Pt_3_Ni@C-Li_2_S_8_ and C-Li_2_S_8_ symmetric battery capacitance curves.

**Figure 5 nanomaterials-11-03416-f005:**
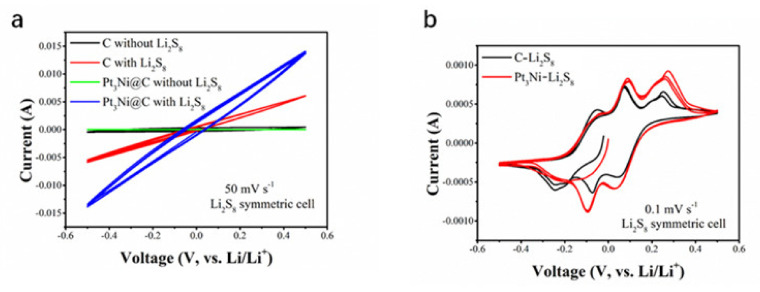
CV curves of the symmetrical cells assembled on two C electrodes with or without Li_2_S_8_ electrolyte and two Pt_3_Ni@C electrodes with or without Li_2_S_8_ electrolyte. (**a**) 50 mV s^−1^. (**b**) 0.1 mV s^−1^.

**Figure 6 nanomaterials-11-03416-f006:**
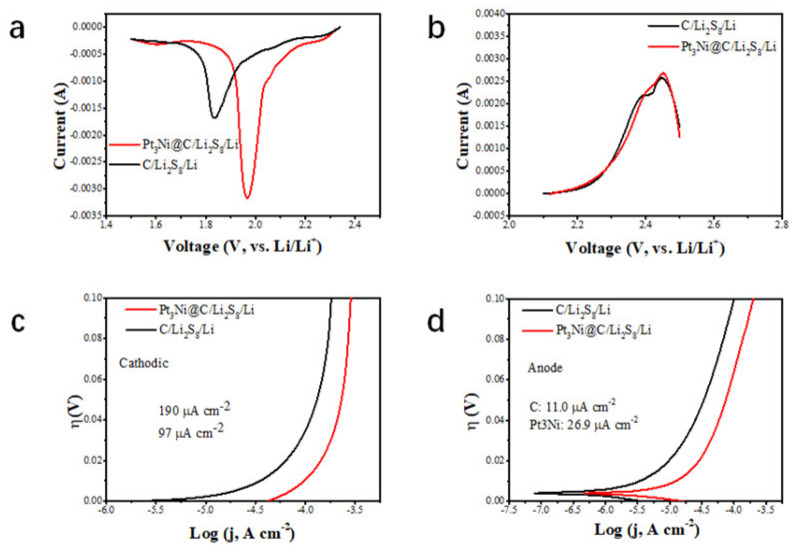
Electrochemical performance of Pt_3_Ni@C-Li_2_S_8_//Li and C-Li_2_S_8_//Li. (**a**) Scanning CV curve of Pt_3_Ni@C-Li_2_S_8_//Li and C-Li_2_S_8_//Li cathode. (**b**) Scanning CV curves of Pt_3_Ni@C-Li_2_S_8_//Li and C-Li_2_S_8_//Li anode. (**c**) Cathodic polarization curves of Pt_3_Ni@C-Li_2_S_8_//Li and C-Li_2_S_8_//Li. (**d**) Anodic polarization curves Pt_3_Ni@C-Li_2_S_8_//Li and C-Li_2_S_8_//Li.

**Figure 7 nanomaterials-11-03416-f007:**
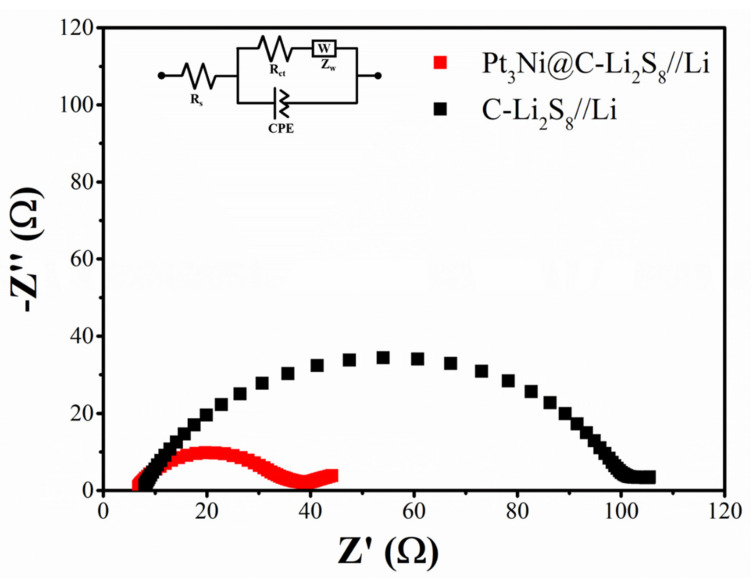
Electrochemical impedance spectroscopy (EIS) of Pt_3_Ni@C-Li_2_S_8_//Li and C-Li_2_S_8_//Li.

## Data Availability

Data are contained within the article.

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
