# Peer review of "Pt3Ni@C Composite Material Designed and Prepared Based on Volcanic Catalytic Curve and Its High-Performance Static Lithium Polysulfide Semiliquid Battery"

_nanomaterials, 2021, doi:10.3390/nano11123416_

Round 1
Reviewer 1 Report
The paper under examination is devoted to obtaining and studying the technical capabilities of galvanic cells, in which lithium polysulfides are used as catholyte, and a composite material containing carbon, platinum and nickel is used as a cathode.
The material presented in the paper is of undoubted academic and practical interest, and its content of this paper is presented in general in good language and style (although I do not presume to judge the quality of the English language used in it, since I do not consider myself competent in this matter). Therefore, I have no objection to its publication in Nanomaterials. Nevertheless, there are some comments and suggestions concerning this paper that should be taken into account before it will be sent to the publication.
1 As far as I understood from the annotation and the text of the article itself, the authors propose to use the polysulfide Li2S2 polysulfide Li2S8, although in principle other polysulfides of this s-element can exist and be used in the experiment, both with a smaller and a more number of sulfur atoms in the molecule from the calculation per lithium atom. It is necessary to somehow explain the choice namely of Li2S8. The aforementioned also applies to the Pt3Ni@C composite, since the reason for its choice as an object of research is also not commented in this paper in any way.
2 Since the paper was submitted to the Nanomaterials, it seems to me that data should be presented in it proving that at least one of the research objects used (either Li2S8 or Pt3Ni@C) consists of nanoparticles. The authors gave in the paper the Fig 2 with the TEM images of Pt3Ni/ C composite material with a magnification 60,000 and 200,000 times, and if so, then in the text of the paper, it is necessary to indicate the size range of those nanoparticles that make up this material.
3 Author Contributions and References list are not drawn up in full compliance with the standards adopted in Nanomaterials journal and, therefore, need to be corrected.
Reviewer 2 Report
In this work, the author prepared a Pt3Ni@C composite as an excellent active material for a static lithium battery. They demonstrated that such Pt3Ni@C-based battery show a good capacity of 550 mAh g-1, and superior cycling performance. I suggest the author carefully solve the following issues,
Some comments are given as follows,
- The scale bar should add in Figure 2.
- There are small mistakes in the main text, please check the main text carefully.
- Some references based on energy harvesting and storage devices are suggested to cite, Microporous and Mesoporous Materials, 2019, 273, 148-155; Nano Energy, 2021, 90, 106536.
- The equivalent circuit diagram should add in Figure 7.
- Did the author check the XRD or TEM of their electrode after the cycling test?
- Why the Pt3Ni@C composite electrode shows good cycling stability?
- Did the author compare their capacity with previous reports?
Round 2
Reviewer 2 Report
The revised manuscript can be accepted.
This manuscript is a resubmission of an earlier submission. The following is a list of the peer review reports and author responses from that submission.
Round 1
Reviewer 1 Report
With the fast development of electric vehicles and devices, Li-S battery is a potential solution for energy density and safety problem of traditional LIBs. However, there are still lots of challenges of Li-S batteries. In this paper, authors developed a cathode material for suppressing the shuttle effect. I would like to recommend the paper to be published in Journal of Electrochem after authors address the following questions:
- Fig 2 needs a scale bar. And do authors have SEM images for the cathode material? From the TEM images, the particle looks like a sphere shape with some agglomeration problem. The morphology is better expressed with SEM images as well.
- Authors got the particle size from TEM. But TEM can only test some particles. Besides, due to the agglomeration problems, the particle size from TEM may not accurate. Do authors have the result of particle size from particle size analyzer? And if authors have BET and pore size results of the cathode material, it will be stronger evidence to support the conclusion.
- What is the ratio of Pt3Ni and C? There may be some loss during the preparation process. Authors should provide the exact number of the C content.
- Could authors provide the loading of the electrode?
- The efficiency is lower than 100% during cycling. Please explain why.
- The capacity retention of Pt3Ni@C is faster than C after 100 cycles. Are there any assumptions for this problem?
- The manuscript has some format problems and typo. Please corrected them. For example, line 236 has an ‘a’ between ‘prepared’ and ‘to’, which should be a typo. Experimental section has some chemical name without right subscript.